# Identification and Characterization of *Neofusicoccum stellenboschiana* in Branch and Twig Dieback-Affected Olive Trees in Italy and Comparative Pathogenicity with *N. mediterraneum*

**DOI:** 10.3390/jof9030292

**Published:** 2023-02-23

**Authors:** Giuliano Manetti, Angela Brunetti, Valentina Lumia, Lorenzo Sciarroni, Paolo Marangi, Nicola Cristella, Francesco Faggioli, Massimo Reverberi, Marco Scortichini, Massimo Pilotti

**Affiliations:** 1Research Centre for Plant Protection and Certification (CREA-DC), Council for Agricultural Research and Economics, 00156 Rome, Italy; 2Terranostra S.r.l.s., 72021 Francavilla Fontana, Italy; 3Department of Environmental Biology, University Sapienza, 00165 Rome, Italy; 4Research Centre for Olive, Fruit trees and Citrus Crops, Council for Agricultural Research and Economics (CREA) (CREA-OFA), 00134 Rome, Italy

**Keywords:** olive tree, *Botryosphaeriaceae*, *Neofusicoccum stellenboschiana*, *Neofusicoccum mediterraneum*, branch and twig dieback, BTD, *Xylella fastidiosa*, olive quick decline syndrome, OQDS, wilting, canker

## Abstract

For about a decade, olive groves in Apulia (Southern Italy) have been progressively destroyed by Olive Quick Decline Syndrome (OQDS), a disease caused by the bacterium *Xylella fastidiosa* subsp. *pauca* (Xfp). Recently, we described an additional wilting syndrome affecting olive trees in that area. The botryosphaeriaceous fungus *Neofusicoccum mediterraneum* was found associated with the diseased trees, and its high virulence toward olive trees was demonstrated. Given the common features with Branch and Twig Dieback (BTD) of olive tree, occurring in Spain and California, we suggested that the observed syndrome was BTD. During our first survey, we also found a botryosphaeriaceous species other than *N. mediterraneum*. In the present article, we report the morphological and molecular characterization of this fungal species which we identified as *Neofusicoccum stellenboschiana*. In the study, we also included for comparison additional *N. stellenboschiana* isolates obtained from olive trees in Latium and Tuscany region (Central Italy). The occurrence of *N. stellenboschiana* in olive trees is reported here for the first time in the northern hemisphere. The pathogenicity and virulence were tested in nine inoculation trials, where the Apulian *N. stellenboschiana* isolate was compared with the isolate from Latium and with the Apulian isolate of *N. mediterraneum*. Both isolates of *N. stellenboschiana* proved pathogenic to olive trees. They caused evident bark canker and wood discolouration when inoculated at the base of the stem of two/three-year-old trees and on one-year-old twigs. However, virulence of *N. stellenboschiana* was significantly lower, though still remarkable, compared with *N. mediterraneum* in term of necrosis progression in the bark and the wood and capacity of wilting the twigs. Virulence of *N. stellenboschiana* and *N. mediterraneum* did not substantially change when inoculations were performed in spring/summer and in autumn, suggesting that these fungal species have the potential to infect and damage olive trees in all seasons. The high thermotolerance of *N. stellenboschiana* was also revealed with in vitro growth and survival tests. The high virulence of these *Botryosphaeriaceae* species highlights their contribution in BTD aetiology and the necessity to investigate right away their diffusion and, possibly, the role of additional factors other than Xfp in the general decline of olive groves in Apulia. Hence the importance of assessing the degree of overlap of BTD/Botryosphariaceae with OQDS/Xfp is discussed.

## 1. Introduction

Since its origin in the Mediterranean basin, approximately 6000–8000 years ago [1], and until the present day, the olive tree (*Olea europaea* L.) has always been a key crop for human settlements and society in this world area due to the nutritional and nutraceutical value of olives and oil. However, its importance goes even further as its cultivation promotes landscape enhancement and the recovery of marginal areas that would otherwise go unused [2,3,4,5].

In Apulia (Southern Italy), the olive tree has always been widespread, with large trees characterizing and enriching the landscapes. Regrettably, the appearance and spread in the Salento peninsula of Olive Quick Decline Syndrome (OQDS) incited by the bacterium *Xylella fastidiosa* subsp. *pauca* (Schaad, Postnikova, Lacy, Fatmic, and Chang) (Xfp), caused the complete destruction of many olive groves so much as to change the very characteristics of the Apulian environment. By inhabiting the xylem, Xfp heavily impacts the normal function of water conduction in the vessels, thus causing rapid canopy scorching and the death of the trees [6,7]. Moreover, edaphic and climate factors might predispose the trees to bacterial pathogenicity, thus acting as disease enhancers [8].

Interestingly, since the OQDS outbreak occurred, other decline syndromes of olive trees characterized by a general dieback, canker and death of branches and twig wilting were reported in Italy (including Apulia) and other parts of the world. These were associated with a mycobiota in which some fungal taxa played major roles—*Botryosphaeriaceae*, *Pleurostomophora richardsiae*, *Phaeoacremonium* spp., *Cytospora* spp., Phaeomoniellales, *Arthrinium marii*, *Neofabraea kienholzii*, and *Phlyctema vagabunda* [9,10,11,12,13,14,15,16,17,18,19,20,21].

Since summer 2019, a wilting syndrome affecting olive trees in the provinces of Brindisi and Taranto has been noted during our surveys in Apulia and suspected of being different from the Xfp-incited OQDS that was also present in those areas. Symptomatology was characterized by red-bronze necrotic patches scattered on the leaf blade or starting from the leaf edge or the main vein. The necrotic lesions tended to spread and coalesce, thus affecting the entire leaf blade, which eventually rolled downward and wilted. A generalized chlorosis of foliage might precede this symptom progression. A discolouration pattern, wedge-shaped when seen in cross section and in the most evident cases, was also present in the xylem as well as sunken, reddish, necrotized bark areas (Figure A1).

We thus began to investigate the nature of this syndrome by assessing the occurrence of potentially pathogenic fungi. After running a preliminary sampling, we isolated some botryosphaeriaceous fungi. One of these was identified as *Neofusicoccum mediterraneum* Crous, M.J. Wingf., and A.J.L. Phillips, a fungal species previously reported in California and Spain as the causal agent of severe Branch and Twig Dieback (BTD) of olive trees [9,11,13]. Indeed, the pathogenicity trials performed with the Apulian isolate clearly showed the high aggressiveness of this species and its capacity to reproduce the observed symptomatology, confirming what had been previously reported for this species [11,13,22].

However, other botryosphaeriaceous fungi were also isolated in our first survey hypothesizing a complex aetiology for the observed syndrome (i.e., based on a multiplicity of *Botryosphaeriaceae* species), similarly to what has been reported, for example, in grapevines [23], olives [9,11,13], and oleasters [18]. Hence, the aim of this study was: (i) to characterize these additional botryosphaeriaceous fungi, which were identified as *Neofusicoccum stellenboschiana* (Tao Yang and Crous), also including isolates of the same species, which had been contemporaneously found in Latium and Tuscany on olive trees; (ii) to assess the pathogenicity and virulence degree of the Apulian isolate of *N. stellenboschiana*, comparing with the isolate of the same species from the Latium region and with the Apulian *N. mediterraneum* isolate that has been the subject of a previous report [22].

## 2. Materials and Methods

### 2.1. Fungal Isolates

Previously, during a survey in an olive orchard located in the municipality of Mesagne (Brindisi province, Apulia, southern Italy), we found olive trees severely affected by BTD. Regarding this phytopathological case, we published a first study focused on the characterization of *N. mediterraneum*, a botryosphaeriaceous fungus that turned out to be strongly involved in the disease [22]. In the present study, we focused on additional *Botryosphaeriaceae*-like isolates, different from *N. mediterraneum*, which were also found associated with BTD in that survey (CREA-DC TPR OL. 431, 438). Given the morphological resemblance with these isolates, we also included in the study two *Botryosphaeriaceae*-like isolates which we found in the Latium region in a young olive groove suffering a lethal wilting by *Verticillium dahliae* (CREA-DC TPR OL.60) and in Tuscany in association with a single olive tree standing in a condominium garden and severely affected by dieback namely severe death of branches and stem cankers (CREA-DC TPR OL.453). Pathogenicity trials were performed to characterize the virulence of the Apulian and Latium isolates and clarify the link of the Apulian isolate with BTD. Importantly, this study was also conducted by simultaneously comparing with the Apulian *N. mediterraneum* (CREA-DC TPR OL.427). A part of the results regarding *N. mediterraneum* have been previously published, as detailed below.

The survey, sample collection, isolations, and concomitant detection of *Xylella fastidiosa* subsp. *pauca* are described in Brunetti et al. [22]. All the activities developed for each isolate are summarized in Table 1.

### 2.2. Morphological Features and Cultural Characteristics of the Selected Fungal Isolates

Conidiomata formation and sporulation were induced on 2% water agar (WA) (Oxoid) with autoclaved *Pinus pinea* needles as the substratum at 23 °C under near-UV light [24]. Fungal structures were observed under a light microscope—a Leica DM6B, equipped with Leica LAS X software, version 3.4.2—in order to record magnified depictions, and a Leica DFC 7000T camera for image acquisition. Conidia germination was induced in microscopic preparations (VEMI S.R.L. Milan, Italy) set up in water and kept at room temperature.

Colonies grown on PDA plates (90 mm in diameter) were observed and photographed at different times in order to document age-related changes.

To determine the optimal growth temperature and the cardinal temperatures for the viability of the isolates, the growth rate and viability were assessed at temperatures ranging from 5 to 45 °C with a step size of five, in the dark. For each temperature, five PDA plates, 90 mm in diameter, were inoculated with a fungal plug 9 mm in diameter taken from an actively growing colony. The diameter of each colony was measured twice at right angles after the colony had covered roughly 70% of the surface for the fast-growing rates (from 15 to 30 °C), or after 12, 8, and 5 days for 5, 10, and 35 °C, respectively. At 40 and 45 °C, fifteen fungal plugs were tested in three plates for each temperature and incubated for 5 days. At 45 °C, plugs were collected from five and nine-day-old colonies in order to verify if survival of fungi was age-related (the test was performed twice). After the incubation period, in case no growth occurred, the viability of the mycelium was tested by transferring the inoculated plates to 25 °C in order to record active growth or the absence of growth (viability vs. death). Growth data were used to infer the medium daily growth rate.

Fungal viability was also tested at 50 °C: plugs were collected from actively expanding colonies (5-day-old) and mature colonies (18-day-old) and incubated at 50 °C in the dark for 6, 18, and 24 h. A total of 15 plugs were used in three PDA plates per treatment (temperature/colony age). Viability was checked as described above.

To assess the growth rate of the botryosphaeriaceous Apulian isolate CREA-DC TPR OL.431 and *N. mediterraneum* (CREA-DC TPR OL. 427) in the regime of high summer temperatures occurring during 2022, we compared their growth rate at 30 °C with that obtained by exposing the cultures to the natural daily temperature variation in the external environment. Outdoor cultures were protected from direct sunlight by keeping them under a portico with natural shadow conditions. All cultures were covered with aluminum foil. The test design was as described above for the determination of optimal growth. The temperature of the outdoor test was monitored on a 30 min basis throughout the duration of the test using a data logger (Cryopak Escort iMINI Temperature Data Logger, Sydney, Australia).

### 2.3. Sequencing and Phylogenetic Analysis for Species Identification

The botryosphaeriaceous isolates were subjected to multi-locus sequencing. After extracting fungal genomic DNA (gDNA) from axenic fungal cultures, the complete ITS region, beta-tubulin 2 (TUB2), translation elongation factor 1-alpha (TEF1-α) and DNA-directed RNA polymerase II second largest subunit (RPB2) were amplified and sequenced to find matches in the NCBI GenBank and to infer a multi-locus phylogeny for species determination. All details on fungal gDNA extraction, primers, reaction assembly, and thermal cycling of the amplifications are reported in Text Appendix A and references therein [25,26,27,28,29].

PCR amplicons were directly sequenced in both directions by Sanger technology (Bio-Fab Research s.r.l., Rome, Italy). Sequences newly generated in this study were deposited in the NCBI GenBank with Acc Nos OP893662-OP893665 (ITS) and OQ091952-63 (TEF1-alpha, TUB2, and RPB2).

All nucleotide sequences generated in this study (i.e., the Italian isolates of *N. stellenboschiana*) are also reported in Text Appendix A. They were compared with GenBank accessions using the blastn suite on the NCBI server (https://blast.ncbi.nlm.nih.gov/Blast.cgi?LINK_LOC=blasthome&PAGE_TYPE=BlastSearch&PROGRAM=blastn) (accessed on 20 September 2022).

Based on the blast results, a *Neofusicoccum*-specific phylogeny was inferred to confirm the specific nature of the isolates under study. A combined multiloci dataset was used as input for the analysis: ITS + TEF1-alpha + TUB2 + RPB2. The reference sequences were extracted from NCBI GenBank. Recent taxonomic studies such as those of Lopes et al. [30], Yang et al. [31], and Zhang et al. [32] were considered models to verify the reliability of our analyses. Generally, all reference species included in the analyses were represented by strains with all four sequence loci, except for a few of them for which strains with the full sequence complement were not found. *Diplodia seriata* and *Phaeosphaeria ammophilae* were included as the outgroups. The complete list and details of the reference sequences used in the phylogenetic analyses are presented in Appendix A.

Sequences of the isolates under study were aligned to reference sequences, each locus separately, with MAFFT on the EMBL-EBI server (htpps://www.ebi.ac.uk) (accessed on 22 September 2022). The alignments were then trimmed to span the same region and concatenated in MEGA X [33]. The evolutionary history was inferred by using the Maximum Likelihood (ML) method and a general time-reversible model [34]. Bootstrap values were calculated from 1000 replicates.

Initial tree(s) for the heuristic search were obtained automatically by applying neighbor-join and BioNJ algorithms to a matrix of pairwise distances estimated using the maximum composite likelihood (MCL) approach, and then selecting the topology with a superior log likelihood value. A discrete Gamma distribution was used to model evolutionary rate differences among sites (5 categories (+*G*, parameter = 0.2528)). Evolutionary analyses were conducted in MEGA X [33].

### 2.4. Pathogenicity Tests

Inoculation trials were carried out to assess the pathogenicity of the botryospaeriaceous isolates under study (CREA-DC TPR OL.431 and 60, from Apulia and Latium, respectively). A direct comparison was also conducted with *N. mediterraneum* (CREA-DC TPR OL427). Olive trees (cv. Leccino and Frantoio) were supplied by Spoolivi-Società Pesciatina d’Olivicoltura (Pescia, PT, Tuscany, Italy), transplanted into 23-L pots, and used for the inoculation trials. The pot substrate was Radicom (Vigorplant, Lodi, Italy), which contained a mixture of peat moss, black peat, marsh peat, and vegetable compost. Humus was present in the form of humic and fulvic acids with a water pH of 6–6.5.

Two/three-year-old olive trees at third/fourth-year-age were used for inoculation of the basal portion of the stem at a height of 15–20 cm and, separately, of one-year-old twigs. For each trial, we used 10 plant replicates for fungal inoculation and five for inoculation with sterile PDA (the negative control). With regard to the twig trial, we inoculated two twigs for each plant replicate both in the fungal and control treatments, thus providing 20 replicates for the fungal-inoculated plot and 10 for the control. The inoculations were performed as follows: rectangular PDA-mycelium plugs (14–16 × 3–4 mm sized for twig, and 18–20 × 8–10 mm for stem inoculation) were cut from an actively growing colony and placed on similarly sized wounds, i.e., on the xylem surface, which had been exposed by cutting the bark top-down with a razor blade through the cambium. The bark strip was then gently set on the plug. The inoculation point was covered with a sterile cotton disk wetted with 3 mL of sterile water and wrapped with a sterile strip of aluminium foil, which was taped to the stem at the top and bottom edges. The cover was removed after 25–30 days. In Table 2, we report details of all pathogenicity trials performed between 2019 and 2021.

The status of the plants was monitored for four to eighteen months or until the death of the inoculated parts.

At the end of the trials, the lengths of the external bark necrosis/canker and internal wood discoloration were recorded. The length of the external healing reactions and of the corresponding wood discolouration behind them was also recorded in the control plants. In the stem-inoculated plants, we calculated the girdling index as the ratio between the tangential spread of bark necrosis and the stem circumference [22]. Eighteen wood fragments were collected from each fungus and sterile PDA-inoculated tree and used to perform fungal re-isolation on streptomycin-supplemented PDA plates.

### 2.5. Statistical Analyses

A one-way, fully randomized analysis of variance (ANOVA or Welch test) and the Tukey test as post-hoc analysis, were carried out to compare the fungal growth at different temperatures, the discolouration streaks on inoculated trees, and the values of the girdling index. PAST version 4.10 was used for the analysis [35].

## 3. Results

### 3.1. Morphocultural Characterization of the Botryosphaeriaceous Isolates

Colonies grown in axenic culture on PDA were moderately fluffy and dirty white in the first four days of growth in the dark, then assumed a green/olivaceous/grey tonality in the central area, sometimes with yellowish nuances. Over time, the whole colony became grey, with darker areas alternating with clearer ones, and areas with fluffy mycelium alternating with others with sparse or flat mycelium (Figure 1).

After seven days of culturing on pine needle agar, isolates produced conidiomata pycnidial, globose, blackish, solitary, or in multilocular stromata embedded in needle tissue and coated with hairy mycelium (Figure 2).

Over time, conidia were observed occasionally in all four isolates by breaking conidiomata on a microscope slide or in mucoid exudates.

Conidia were lacking any persistent mucous sheath; they were hyaline, smooth, thin-walled, ellipsoidal, widest in the middle or in the upper third, apex subobtuse, base subtruncate. Septa were not observed. The mean size of conidia was in the range 20.9–24.2 (length), 6.1–6.9 (width) μm; length/width ratio ranged from 3.2 to 3.7. See Table 3 for the conidia size referred to each isolate and Figure 3 for the images.

We observed germination from a few hours up to 24 h after setting up the microscopic preparations in moist chamber. Germ tubes emerged from one or both of the apical ends or along the conidium side. Up to three germ tubes were observed simultaneously for each germinating conidium. Germinating conidia showed from none up to three septa, most frequently one or two (Figure 3). Microscopic features are consistent with those reported by Yang et al. [31] and Guarnaccia et al. [36] for *N. stellenboschiana*, even though we observed a range for the number of septa in the germinating conidia (0–3) instead of exactly two as previously reported [31,36]. In addition, the conidia of our isolates seem slightly larger than those described by these authors (19–21 × 5.5–6 µm, [31] and 18–21.5 × 4–6.7 μm [36]).

Isolates from Apulia, Latium, and Tuscany (CREA-DC TPR OL.431, 60, and 453) were able to grow at 10–30 °C with a peak at 30 °C. A minimal growth at 5 and 35 °C was also recorded for CREA-DC TPR OL.60 and 453, respectively. Isolates remained viable after a five-day exposure up to 40 °C. At 45 °C, all isolates lost viability when fungal plugs were collected from five-day-old colonies, whereas they partially survived when they were collected from nine-day-old colonies (CREA-DC TPR OL.431, viable 9/13 out of 15 plugs; OL. 60, viable 6/9 out of 15 plugs; and OL.453, viable 10/15 out of 15 plugs; results from two trials), (Figure 4). The growth rate of isolate CREA-DC TPR OL.431 was significantly higher than CREA-DC TPR OL.60 and 453 at all tested temperatures in the range 10–30 °C, except for 10 and 25 °C (*p* < 0.01) (Figure 4). CREA-DC TPR OL.60 and 453 were not statistically different except at 5 and 35 °C. Interestingly, in the survival test at 50 °C, all mycelial plugs taken from both young and mature mycelium (i.e., 5- and 18-day-old cultures) kept their viability at all tested times (i.e., 6, 18, and 24 h). Just one plug out of fifteen-from those taken from five-day-old culture of CREA-DC TPR OL. 453–lost the viability after a 24 h incubation. We did not test longer incubation times because, at 50 °C, the PDA dehydrated and split.

The growth rate at a natural daily temperature regime typical of summer 2022 was very similar to the optimal growth rate at 30 °C for both, *N. stellenboschiana* (CREA-DC TPR OL. 431) and *N. mediterraneum* (CREA-DC TPR OL. 427) in both trials, even though statistical significances were detected (Figure 5).

### 3.2. Sequencing and Phylogenetic Analysis for Species Identification

In all four *N. stellenboschiana* isolates, CREA-DC TPR OL. 60, 431, 438, and 453, the ITS region, and fragments of TEF1-alpha, TUB2, and RPB2 were sequenced. Multiple sequence alignments showed that all four isolates have identical sequences for each locus except for one single nucleotide polymorphism (SNP) in RPB2, which distinguished CREA-DC TPR OL. 438 from the other Italian isolates. Regarding the comparison with reference sequences of known *N. stellenboschiana* strains from the fungal biodiversity center (CBS), nucleotide identity varied in the percentage range 99.6–100% according to the specific pairwise comparison of the various loci in play (isolate under study–CBS strain) (inference made on the spanning region used for phylogenesis, see Appendix A for identity values relating each pairwise comparison).

The phylogenetic relationships of *Neofusicoccum* species fully matched the topology of the *Neofusicoccum* phylogeny reported by Yang et al. [31] and Zhang et al. [32]. Specifically, sequence data of the reference strains of species of the *N. australe* complex grouped together with high statistical support, and each (cryptic) species was clearly distinguished from the others, forming highly supported sub-clades. Sequence data of the *N. stellenboschiana* representatives and the Italian isolates were included without uncertainty in the *N. australe* complex but, coherently with phylogeny by Zhang et al. [32], formed a group with a lower bootstrap value (around 50) compared with the other species of the *N. australe* complex (Figure 6). This phylogenetic feature is shared by all recognized *N. stellenboschiana* strains and, thus, distinguishes this species from the other cryptic species within the complex. Our identification fully matches our actual knowledge on the matter. A complete phylogeny of the *Neofusicoccum* genus is presented in Appendix A and fully confirms the results obtained with the *N. australe* complex-based phylogeny (Figure 6).

### 3.3. Pathogenicity Tests

The inoculation at the basal portion of the stem with *N. stellenboschiana* isolates and *N. mediterraneum* caused severe bark canker that progressed upward and downward from the inoculation point. Evident and even longer discolouration streaks were present in the wood. See figures from 6 to 12 for a complete symptom recording and bar chart elaborations.

Regarding *N. stellenboschiana*, the isolate from Latium caused longer discolouration streaks in October compared with June inoculation (*p* < 0.05) (Figure 7a). A simultaneous comparison between the isolate from Latium and that from Apulia showed that the latter caused longer necrosis streaks, although not with full statistical significance (Figure 7b and Figure 8). On the other hand, in another trial, a simultaneous comparison between *N. stellenboschiana* from Apulia and *N. mediterraneum* from Apulia showed that (i) the progression of necrosis did not significantly differ between June and October inoculations and (ii) necrosis streaks caused by *N. mediterraneum* were always far longer than those caused by *N. stellenboschiana* (*p* < 0.01) (Figure 9, Figure 10 and Figure 11).

The girdling index did not statistically differ among the *N. stellenboschiana* isolates (Figure 7c). On the contrary, it was significantly higher for *N. mediterraneum* compared with the Apulian isolate of *N. stellenboschiana* (*p* < 0.01) (Figure 11b). In fact, *N. stellenboschiana* did not necrotize more than 50–60% of the stem circumference (girdling index was between 0.38 and 0.60), whereas *N. mediterraneum*-caused bark necrosis typically girdled the stem circumference (almost) entirely (girdling index was between 0.62 and 1.00) (Figure 7c and Figure 11b).

When seen in a transversal section, the progression of wood discoloration was wedge/arch-shaped and outside-in. As expected on the basis of the girdling index, *N. stellenboschiana*-caused discolouration affected less than half of the transversal section, whereas *N. mediterraneum*-caused discolouration affected the greater part of the transversal section (Figure 12).

The inoculation of the twigs with the two *N. stellenboschiana* isolates from Apulia and Latium caused evident bark canker and wood discolouration that progressed upward and downward from the inoculation point (Figure 13). The treatments, including the fungal isolate and the inoculation season, did not cause evident differences in the internal/external necrosis lengths. Wilting events were detected with low frequency: isolates from Latium wilted three, two, and five twigs (out of the twenty inoculated in each trial) in the May, July, and October trials, respectively; similarly, isolates from Apulia wilted two, three, and five twigs in the May, July, and October trials, respectively (Figure 14).

A direct comparison of the wilting capacities of *N. stellenboschiana* and *N. mediterraneum* was performed by running a twig-inoculation trial with both fungal species. Twigs inoculated with *N. mediterraneum* were all wilted after 23 days (results published in Brunetti et al. [22]), whereas no wilting was observed in the twigs inoculated with the Apulian *N. stellenboschiana*, even after 10 months of observation, though all inoculated twigs were affected by cankers (Figure 15).

In all plants inoculated both at the stem and the twigs, with *N. stellenboschiana* and *N. mediterraneum* isolates, none or at most slight /moderate or irregular callus reaction were opposed by the host to necrosis progression at the level of the bark.

The inoculation wounds on the twigs and stems of plants used as controls reacted with a strong callus reaction and completely healed. A minimal discolouration was present in the wood behind it. Control plants maintained leaf turgor and growth vigor during the entire observation period. *N. stellenboschiana* and *N*. *mediterraneum* were abundantly reisolated from the spreading necrosis of all the inoculated plants, but not from healthy control plants. Koch’s postulate was thus fulfilled.

## 4. Discussion

Diffusion of *N. stellenboschiana*. The pathosystem *N. stellenboschiana*-olive tree is reported here for the first time in the northern hemisphere. This fungal species was recently described as a result of a multi-locus reclassification of a fungal isolate from grapevine in South Africa [31], previously identified as *N. australe* [37]. Consequently, it was shown that in the same country, olive, fig, peruvian pepper, and apple were also natural hosts for this fungal species, which was associated with evident die-back symptoms and internal wood necroses [38]. At the same time, in the northern hemisphere, *N. stellenboschiana* was reported for the first time to cause cankers on Avocado in Crete (Greece) [36], and involved, along with other *Botryosphaeriaceae*, in a cork oak decline spread throughout the Oran region in Algeria [39].

The fact that our isolates were from Apulia, Latium, and Tuscany clearly suggests that *N. stellenboschiana* has established itself over most of the Italian territory, possibly also colonizing host species other than olive trees, given the host-neutral lifestyle of the *Botryosphaeriaceae* and *N. stellenboschiana* in the specific case [38].

The identification within the *Botryosphaeriaceae*. The multi-locus phylogenetic analysis convincingly identified our isolates as *N. stellenboschiana* and included them in the *N. australe* species complex together with the other members of this group: *N. australe sensu strictu*, *N. luteum*, *N. lumnitzerae*, *N. variabile*, *N. rapanae* and *N. cryptoaustrale* [31,32]. The birth of *N. stellenboschiana* as a new species is emblematic of the continuous taxonomic reassessment that is underway in the *Botryosphaeriaceae*, given the unsuitability of morphological features alone to define taxa. Over time, and especially in recent years, either old species were synonymised or new ones were founded according to a molecular species concept based on multi-locus sequencing and phylogenetic analyses. It is not to be excluded that multi-locus sequencing will enable, in the future, the reclassification of additional isolates previously identified as *N. australe*.

Undoubtedly, the variation (and the similarity at the same time) within a species complex might be typical of cryptic species whose differentiation according to an ecological species concept remains, however, fairly problematic given, for example, the lack of host specialization by several *Botryosphaeriaceae* species [24,30,31,32,40,41,42]. It is important to underline that our *N. stellenboschiana* isolates seem more similar to those described by Guarnaccia et al. from avocado [36] in that the colour rapidly changed from white to grey in 4/5 days, whereas Yang et al. report a dirty white colour [31]. Nevertheless, conidia of Italian isolates resulted slightly larger than those described in both cited studies [31,36]. Given that the foundation of *N. stellenboschiana* is based on just one isolate [31], the phenotypic characterization of the species still awaits completion. Thus, the morphological and physiological features of *N. stellenboschiana* isolates from different host species and world areas would be accurately examined in order to define the intra-specific range of variation and usefully integrate the molecular species concept. Emblematic of the variability within *Botryosphaeriaceae* species is the recent description of a chlamydospore-producing isolate of *N. mediterraneum*, a feature not previously reported in this species [22].

Involvement of *N. stellenboschiana* in the etiology of BTD. In the context of the BTD of olive trees in Apulia [22], *N. stellenboschiana* arises in this work as an additional botryosphaeriaceous agent of this syndrome in line with the fact that *Botryosphaeriaceae* typically co-infect the host leading to disease expression. “Botryosphaeria dieback” of grapevine is the most studied of this taxonomically-related-multi-agent type of disease [23], but several other examples can be cited in which shoot wilting, dieback, branch canker, wood discolouration, and the death of trees are strictly linked to the colonization of diverse pathogenic Botryosphaericeae species, especially those in the genus *Neofusicoccum*, *Diplodia* and *Lasiodiplodia*: decline and mortality of *Eucaliptus camaldulensis* in Sardinia (Italy) [43], dieback and cankers of Loquat [44], decline of almond [45], dieback of lentisk [46], canker and dieback of oleaster [18], branch and twig dieback of olive trees in Spain and California (USA) [11,13], branch cankers and dieback of *Citrus* and *Ficus microcarpa* trees [47,48].

Pathogenicity tests prolonged over several months showed that *N. stellenboschiana* isolates from Apulia and Latium have similar virulence. Inoculations at the base of the stem and on one-year-old twigs always showed an evident capacity for *N. stellenboschiana* and *N. mediterraneum* to cause bark canker and wood discolouration. Interestingly, both fungal species expressed their virulence regardless of the inoculation time (spring/summer and autumn/winter), suggesting that they are fully pathogenic to the olive tree for most of the year in geographical areas characterized by mild winters and hot summers, such as those occurring in Central/Southern Italy. However, though the virulence of *N. stellenboschiana* was noteworthy, *N. mediterraneum* resulted even more virulent in all tested traits of pathogenicity: necrosis progression in the bark and wood, either longitudinally or tangentially (i.e., the capacity of girdling the stem) or transversally (i.e., the capacity of deepening in the xylem) as well as the capacity of wilting the twigs.

All this considered, it would seem that *N. mediterraneum* would play a primary role as an agent of BTD in Apulia compared with *N. stellenboschiana*. Nevertheless, it is worth remembering that some *Botryosphaeriaceae* species usually colonize plants as endophytes or weak pathogens but adopt a pathogenic lifestyle or increase their virulence when environmental stress conditions impact on the host plants [49,50]. Thus, it cannot be excluded that *N. stellenboschiana* is even more virulent than shown in this work when appropriate concurrent stress factors come into play. Artificial infection experiments integrated with abiotic stressors will help to clarify this point.

Beyond the comparison between *N. stellenboschiana* and *N. mediterraneum*, the wood discolouration patterns caused by these species in the inoculation trials (Figure 12) perfectly matched those observed in the olive trees naturally affected by BTD (Figure A1). This also confirms once more the fact that the Koch postulate was fulfilled for both fungal species and demonstrates the causal relationship between the fungal species under study and natural symptoms.

Thermal requirements of *Neofusicoccum* species and adaptation to extreme climate change. Based on growth rates in *in vitro* tests at different temperatures, it seems that isolates from Latium and Tuscany are very similar but distinct from those from Apulia. Interestingly, all three tested isolates resulted in being highly thermotolerant, similarly to *N. mediterraneum* [22], as they were able to survive prolonged exposure at high temperatures, up to 50 °C. Moreover, both *N. stellenboschiana* and *N. mediterraneum* performed a nearly optimal growth at the daily thermal regime occurring in summer 2022, which proved to be the warmest year in Italy since 1800 (https://www.isac.cnr.it/climstor/climate_news.html#year-to-date) (accessed on 22 January 2023). This clearly suggests that these fungal species are well adapted to and emerging in an environment subjected to extreme climate change.

## 5. Conclusions

The first approach to the study of BTD in Apulia opens up new perspectives and raises questions about the decline of olive trees. To note that in our ongoing survey of BTD in Apulia, we are repeatedly detecting *N. mediterraneum* and *N. stellenboschiana*, for example in Nardò (province of Lecce) and Sava (Taranto) as well as additional *Botryosphaeriaceae* species and other potentially pathogenic fungal taxa so that the aetiology of BTD outlines more and more as a complex matter (unpublished data).

Thus, all findings provided to date on BTD in Apulia are the first piece of a puzzle that still awaits completion. Describing the pathogenic microbiota associated with a high number of BTD-affected trees in different areas of Apulia, including the annotation of Xfp presence in the same trees, would now be due and urgent for better comprehending the nature of the syndrome [22,51]. Equally important for this purpose will be to assess whether abiotic stressors, among those acting in the olive groves in Apulia [8], directly or indirectly condition virulence of fungi and resilience of the host plant. In other words, we retain reliable a scenario in which predisposing, inciting, and contributing factors altogether work to determine the collapse of the host immune system, thus establishing a *sensu strictu* decline [52,53]. Under this specific point of view, and based on the fact that *Botryosphaeriaceae* species frequently occur in Xfp-infected trees (unpublished data), we are tempted to speculate on an additional level of interaction. Namely, just assumed that BTD and OQDS in Apulia are two distinct diseases, they might also represent components of the same decline phenomenon when they simultaneously affect the same trees. In this case, a likely scenario, among the many possible ones, might frame Xfp as a predisposing or inciting factor, together with abiotic stressors and *Botryosphaeriaceae* (and possibly other fungal species) as inciting or contributing factors.

## Figures and Tables

**Figure 1 jof-09-00292-f001:**
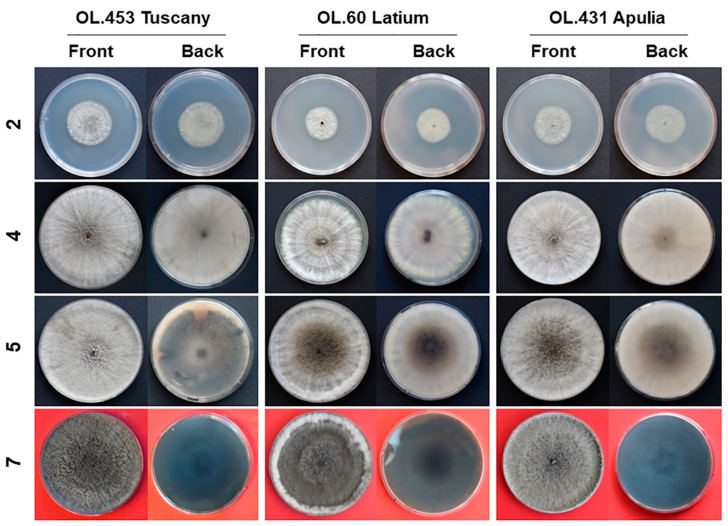
Axenic cultures of *Neofusicoccum stellenboschiana* on PDA (isolates CREA-DC TPR OL.453, 60 and 431). On the left age of the cultures in days.

**Figure 2 jof-09-00292-f002:**
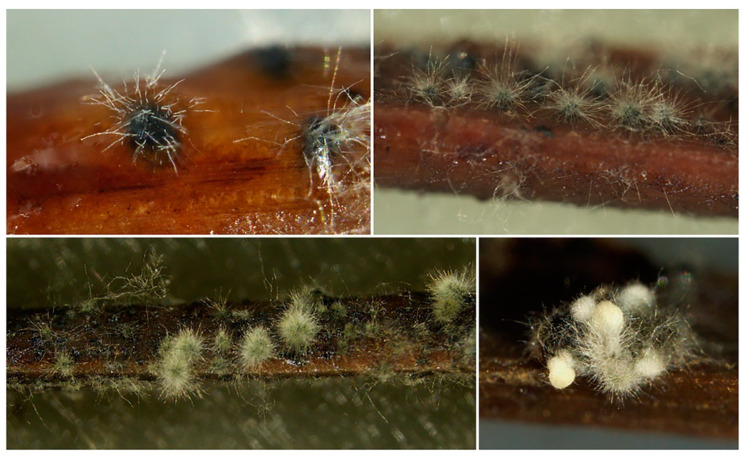
Conidiomata of *Neofusicoccum stellenboschiana* developed on pine needles, progressively coated with hairy mycelium and exuding conidial masses (picture (**below**) and (**right**)).

**Figure 3 jof-09-00292-f003:**
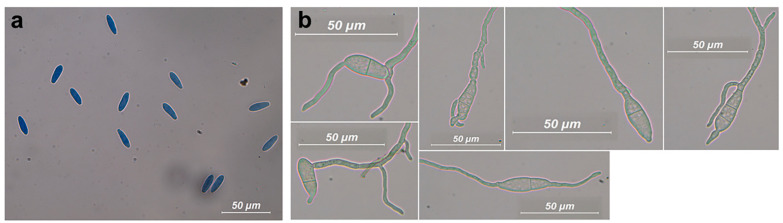
Conidia of *Neofusicoccum stellenboschiana*. (**a**) Conidia were observed after squashing the conidiomata which had been developed on pine needles or in mucoid exudates of conidiomata. (**b**) Germinating conidia with a variable number of septa and germ tubes.

**Figure 4 jof-09-00292-f004:**
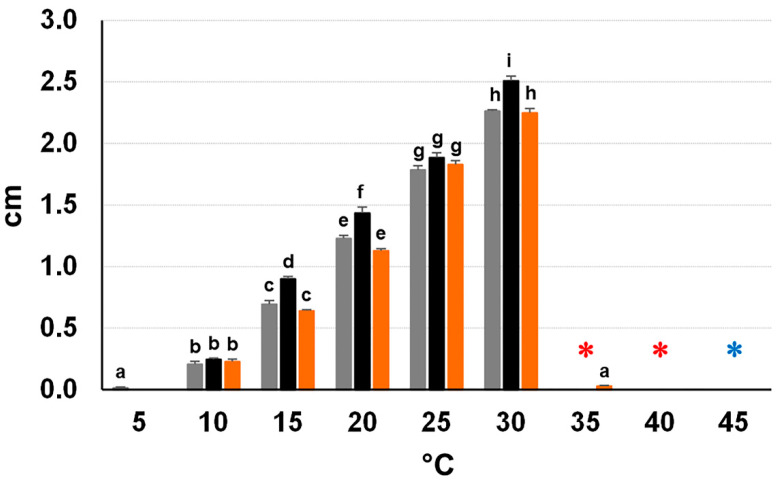
Daily growth rates of *Neofusicoccum stellenboschiana* isolates (CREA-DC TPR OL.60 from Latium, in grey; CREA-DC TPR OL.431 from Apulia, in black; and CREA-DC TPR OL.453 from Tuscany, in orange) on PDA at different temperatures. Different letters indicate statistically significant differences (*p* < 0.01). The bars indicate the standard errors. Red asterisks mean that fungi were still viable after a five-day incubation at 35 and 40 °C. The azure asterisk indicates survival of a variable number of mycelial plugs out of a total of 15 even after a five-day incubation at 45 °C (see the text).

**Figure 5 jof-09-00292-f005:**
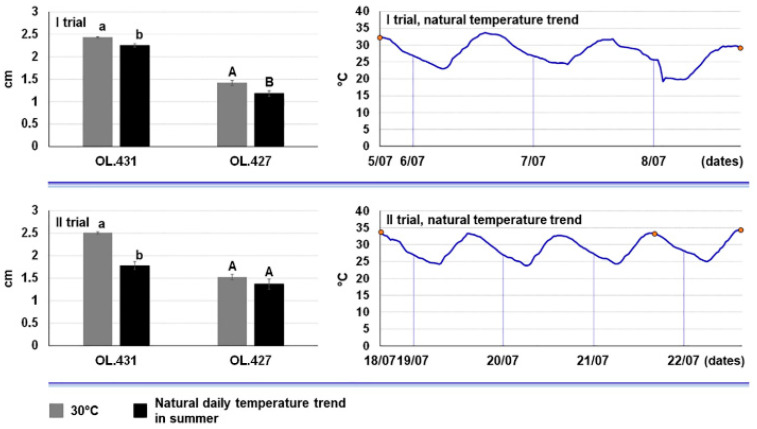
Growth rate of *Neofusicoccum stellenboschiana* (CREA-DC TPR OL.431) and *N. mediterraneum* (CREA-DC TPR OL.427) at 30 °C (optimum temperature in in vitro growth) compared with growth rate of the same fungi under a natural temperature trend occurring in July 2022. Comparisons were statistically analysed for each fungal species separately. Different letters indicate significant differences (*p* < 0.01). The orange dots indicate the beginning and the end of the trial; in the II trial the second dot represents the end of the test with *N. stellenboschiana* and the third dot indicates the end of the test with *Neofusicoccum mediterraneum*.

**Figure 6 jof-09-00292-f006:**
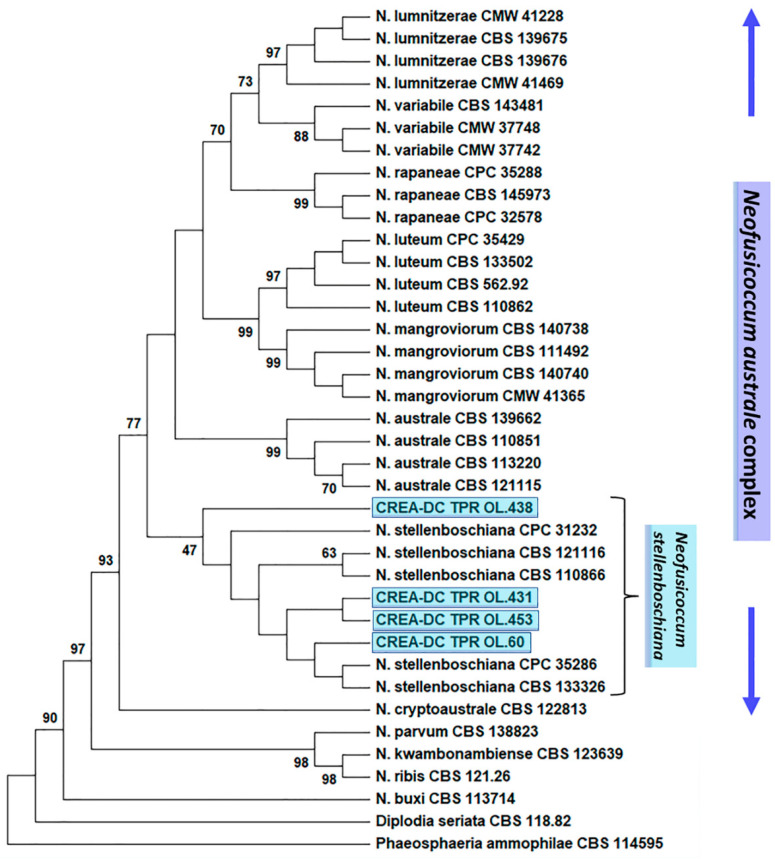
Phylogenetic tree of *Neofusicoccum australe* complex, based on ITS + TEF1-alpha + TUB2 + RPB2 data set and including the botryosphaeriaceous isolates from Italy: CREA-DC TPR OL.431 (Apulia), 438 (Apulia), 60 (Latium), 453 (Tuscany) (shaded in blue-sky). The phylogeny was inferred using the Maximum Likelihood (ML) method and general time reversible model [34], with 1000 bootstraps. The tree with the highest log likelihood (−7572.55) is shown. The percentage of trees in which the associated taxa clustered together is shown next to the branches (values > 40). This analysis involved 38 nucleotide sequences. There were a total of 2367 positions in the final dataset. Evolutionary analyses were conducted in MEGA X [33].

**Figure 7 jof-09-00292-f007:**
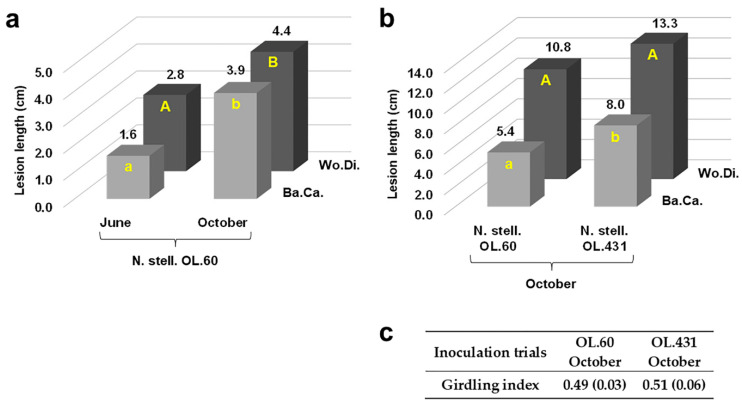
Outcome of inoculations with *Neofusicoccum stellenboschiana* at the base of the stem of three-year-old olive trees in term of lesion length of bark canker (Ba.Ca.) and wood discolouration (Wo.Di.). (**a**) Virulence of *N. stellenboschiana* from Latium (CREA-DC TPR OL.60) in relation to different inoculation times; different letters indicate statistically significant differences (*p* < 0.05). (**b**) Comparison between *N. stellenboschiana* from Latium and from Apulia (CREA-DC TPR OL.431); different letters indicate statistically significant differences (*p* < 0.05). (**c**) Girdling index of Latium and Apulia isolates (no statistically significant differences detected, *p* = 0.435).

**Figure 8 jof-09-00292-f008:**
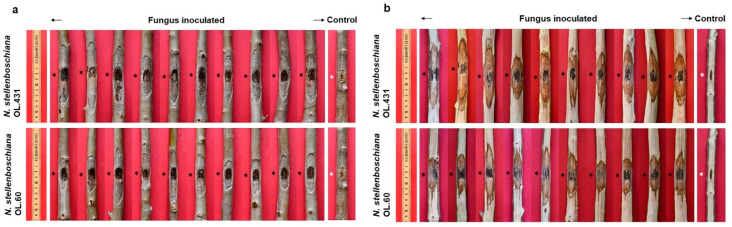
Outcome of inoculation of olive tree cv Frantoio with *Neofusicoccum stellenboschiana* isolates from Apulia (CREA-DC TPR OL.431) and Latium (CREA-DC TPR OL.60). (**a**) Bark cankers. (**b**) Underlying discolouration streaks affecting the wood. Black and white asterisks indicate the inoculation point.

**Figure 9 jof-09-00292-f009:**
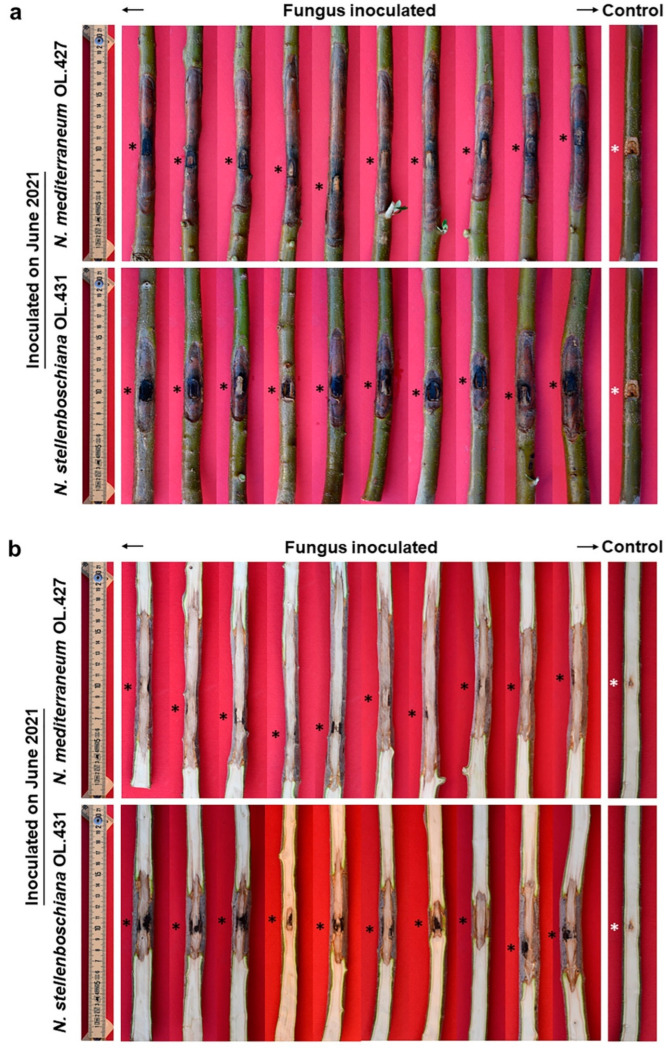
Outcome of June inoculation of olive tree cv Frantoio with *Neofusicoccum stellenboschiana* (CREA-DC TPR OL.431) compared with *N. mediterraneum* (CREA-DC TPR OL.427). (**a**) Bark cankers. (**b**) Underlying discolouration streaks affecting the wood. Black and white asterisks indicate the inoculation point. Pictures of *N. mediterraneum* symptoms have been previously published [22] and are reported here to compare with *N. stellenboschiana*.

**Figure 10 jof-09-00292-f010:**
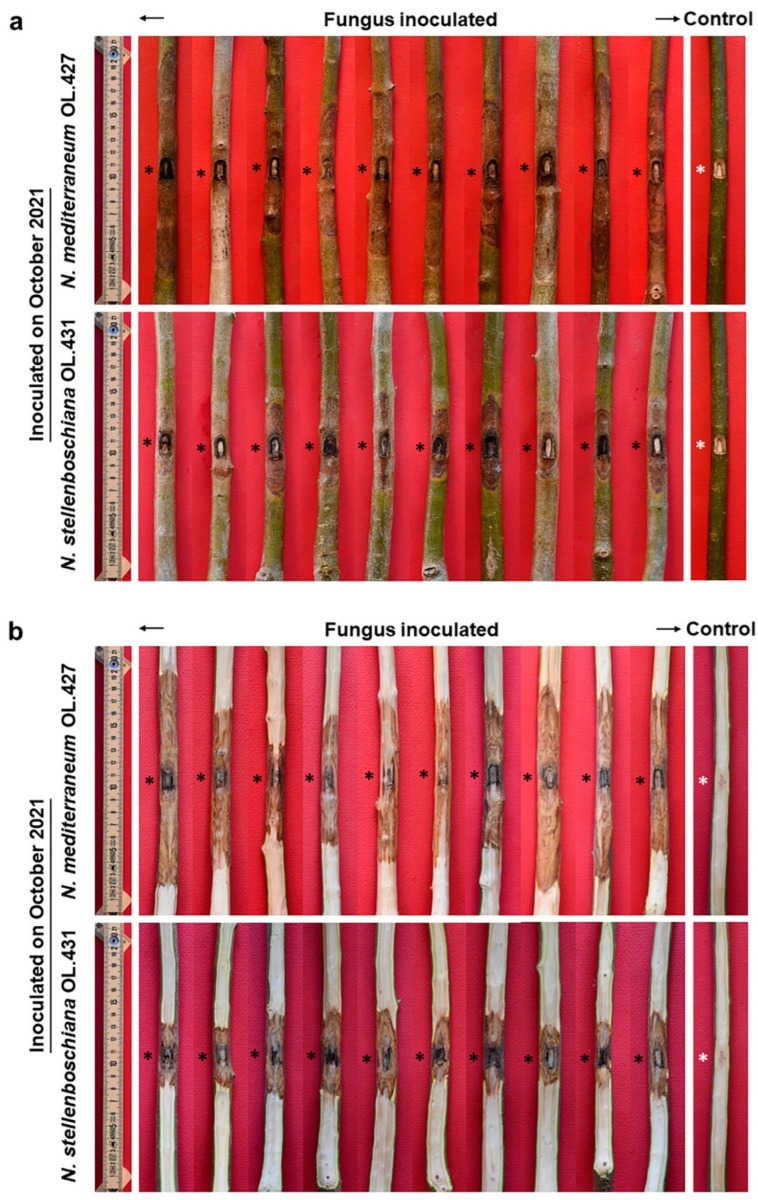
Outcome of October inoculation of olive tree cv Frantoio with *Neofusicoccum stellenboschiana* (CREA-DC TPR OL.431) compared with *N. mediterraneum* (CREA-DC TPR OL.427). (**a**) Bark cankers. (**b**) Underlying discolouration streaks affecting the wood. Black and white asterisks indicate the inoculation point.

**Figure 11 jof-09-00292-f011:**
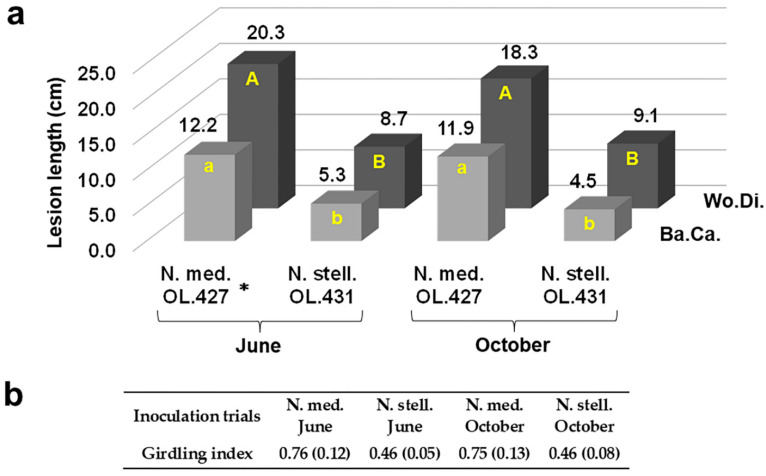
Outcome of inoculations at the base of the stem of three-year-old olive trees to compare virulence of *Neofusicoccum stellenboschiana* (CREA-DC TPR OL.431) and *N. mediterraneum* (CREA-DC TPR OL.427). (**a**) Direct comparison between the two fungal species also in relation to different inoculation times (Ba.Ca. = Bark Canker, Wo.Di. = Wood Discolouration); different letters indicate statistically significant differences (*p* < 0.01). (**b**) Girdling index: differences were significant between the two fungal species within each and regardless of the inoculation time (*p* < 0.01). * Data regarding June inoculation of *N. mediterraneum* have been previously published [22] and are reported here to compare with *N. stellenboschiana*.

**Figure 12 jof-09-00292-f012:**
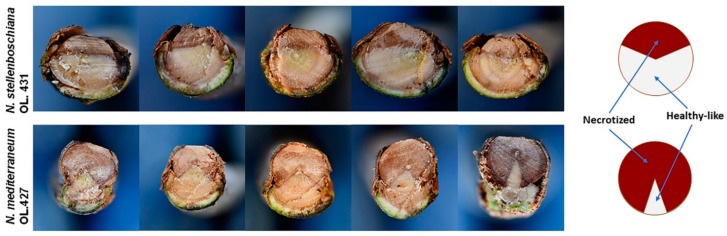
Representative cases of girdling necrosis and its progression in the xylem according to a wedge/arch-shaped model (transversal section) following inoculation at the base of the stem of three-year-old olive trees with *Neofusicoccum stellenboschiana* (CREA-DC TPR OL.431) and *N. mediterraneum* (CREA-DC TPR OL.427). Inoculation performed on 17 June 2021 (see Table 2). Note that girdling necrosis and wedge-shaped discolouration are less extended in *N. stellenboschiana*-inoculated trees than in those inoculated with *N. mediterraneum*. Images relative to *N. mediterraneum* have been previously published by Brunetti et al. [22] and are included for comparison. On the right a model of the necrosis progression pattern typical of *N. stellenboschiana* (**above**) and *N. mediterraneum* (**below**).

**Figure 13 jof-09-00292-f013:**
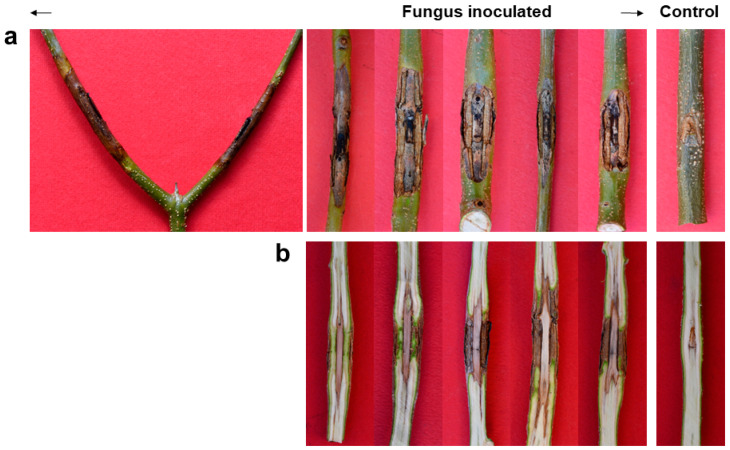
Outcome of inoculation of one-year-old twigs of olive trees with *Neofusicoccum stellenboschiana.* (**a**) Bark cankers. (**b**) Underlying discolouration streaks affecting the wood. Some representative inoculations are depicted.

**Figure 14 jof-09-00292-f014:**
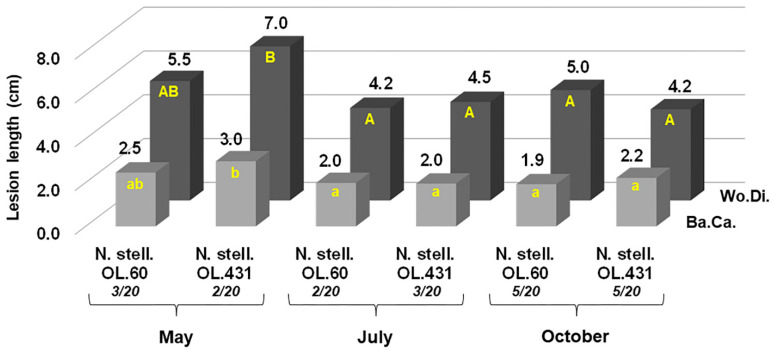
Outcome of inoculations on one-year-old twigs of olive trees to compare virulence of the *Neofusicoccum stellenboschiana* isolates from Latium and Apulia (CREA-DC TPR OL.60; CREA-DC TPR OL.431) also in relation to three different inoculation times (Ba.Ca. = Bark Canker, Wo.Di. = Wood Discolouration); different letters indicate statistically significant differences (*p* < 0.05). The numerical rates in italics under the isolate codes indicate the number of wilted twigs out of twenty which were inoculated, over an observation period of eight months.

**Figure 15 jof-09-00292-f015:**
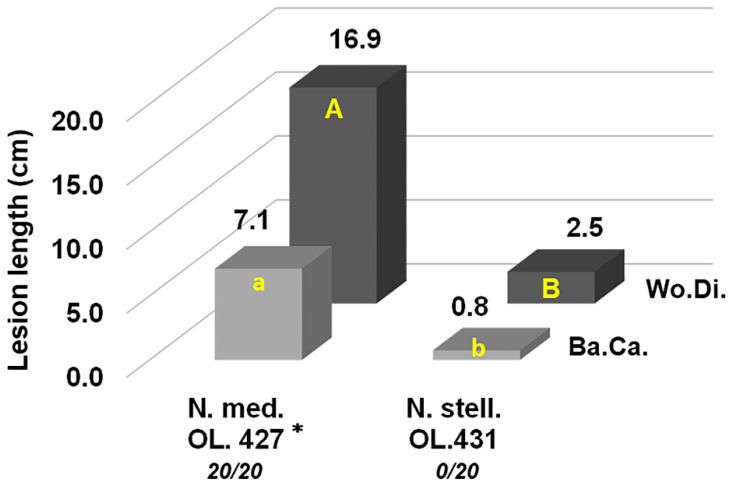
Outcome of inoculations on one-year-old twigs of olive trees to compare virulence of the *Neofusicoccum stellenboschiana* isolate from Apulia (CREA-DC TPR OL.431) and *N. mediterraneum* (CREA-DC TPR OL.427) (Ba.Ca. = Bark Canker, Wo.Di. = Wood Discolouration); different letters indicate statistically significant differences (*p* < 0.01). The numerical rates in italics under the isolate codes indicate the number of wilted twigs out of twenty which were inoculated over an observation period of ten months. Importantly *N. mediterraneum* wilted all the inoculated twigs within twenty-three days after the inoculation. * Data of *N. mediterraneum* inoculation have been previously published [22] and are reported here to compare with *N. stellenboschiana*.

**Table 1 jof-09-00292-t001:** Characterization of Botryosphaeriaceous isolates from olive tree in Italy. The symbol “✓” means that the activity has been performed/fulfilled in this work.

Fungal Isolates under Study	Abbreviated Code * (Origin)	Multi-Locus Sequencing	Microscopic Observations	In Vitro Growth	In Vitro Growth at 30 °C vs. Daily Summer Temperatures	Pathogenicity Trials
*Neofusicoccum mediterraneum*	OL.427 (Apulia)	[22]	[22]	[22]	✓	[22] and ✓
*Neofusicoccum* sp. 1	OL.431 (Apulia)	✓	✓	✓	✓	✓
*Neofusicoccum* sp. 2	OL.438 (Apulia)	✓	✓			
*Neofusicoccum* sp. 3	OL.60 (Latium)	✓	✓	✓		✓
*Neofusicoccum* sp. 4	OL.453 (Tuscany)	✓	✓	✓		

* The full name of the code implies adding “CREA-DC TPR” before each abbreviated code.

**Table 2 jof-09-00292-t002:** Pathogenicity trials performed on olive tree with *Neofusicoccum stellenboschiana* and, for comparison, *N. mediterraneum*.

Pathogenicity Trials	Cultivar	Age of the Trees (Year)	Inoculation Time	Duration in Months (mo)	Average Diameter at the Inoculation Point
**Stem trials**					
*N. stellenboschiana* OL.60	Leccino	2/3	20 June 2019	12 mo	1.12
*N. stellenboschiana* OL.60	Leccino	2/3	2 October 2019	12 mo	1.70
*N. stellenboschiana* OL.60 vs. *N. stellenboschiana* OL.431	Frantoio	3/4	15 October 2020	18 mo	1.76
*N. stellenboschiana* OL.431 vs. *N. mediterraneum* OL.427 *	Frantoio	3/4	17 June 2021	4 mo	1.50
*N. stellenboschiana* OL.431 vs. *N. mediterraneum* OL.427	Frantoio	3/4	18 October 2021	4 mo	1.56
**Twig trials**					
*N. stellenboschiana* OL.60 vs. *N. stellenboschiana* OL.431	Frantoio	2/3	15 May 2020	8 mo	0.52
*N. stellenboschiana* OL.60 vs. *N. stellenboschiana* OL.431	Frantoio	2/3	15 July 2020	8 mo	0.51
*N. stellenboschiana* OL.60 vs. *N. stellenboschiana* OL.431	Frantoio	2/3	14 October 2020	8 mo	0.54
*N. stellenboschiana* OL.431 vs. *N. mediterraneum* OL.427 *	Frantoio	2/3	17 May 2021	10 mo	0.41

* Results of *N. mediterraneum* inoculations have been previously published [22] and are reported here to compare with *N. stellenboschiana.*

**Table 3 jof-09-00292-t003:** Conidia size of Italian *Neofusicoccum stellenboschiana* isolates from olive trees.

*Neofusicoccum stellenboschiana* Isolate	Length * Mean (Range, SD) (µm)	Width * Mean (Range, SD) (µm)	Length/Width
CREA-DC TPR OL.431 (Apulia)	22.6 (19.1–25.0, 1.3)	6.1 (5.0–6.7, 0.4)	3.7
CREA-DC TPR OL.438 (Apulia)	23.3 (18.3–29.4, 2.5)	6.9 (6.2–7.6, 0.3)	3.4
CREA-DC TPR OL.60 (Latium)	24.2 (19.3–29.0, 2.0)	6.8 (5.4–8.2, 0.7)	3.6
CREA-DC TPR OL.453 (Tuscany)	20.9 (18.2–24.5, 1.1)	6.6 (5.7–8.3, 0.5)	3.2

* Fifty conidia were measured for each isolate.

## Data Availability

Not applicable.

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
