# Peer review of "Identification and Characterization of Neofusicoccum stellenboschiana in Branch and Twig Dieback-Affected Olive Trees in Italy and Comparative Pathogenicity with N. mediterraneum"

_jof, 2023, doi:10.3390/jof9030292_

Round 1
Reviewer 1 Report
Please see the file attached

Reviewer 2 Report
Dear editor:
The manuscript entitled "First report of Neofusicoccum stellenboschiana in Branch and Twig Dieback-affected Olive Trees in the Northern Hemi- 3 sphere (Salento, Apulia, Italy) and comparison of Pathogenicity 4 with N. mediterraneum" has been reviewed. The causal pathogen was well identified. However, there some issues need to be revised.
1. The title is better to be reconsidered. How about delete ‘and comparison of Pathogenicity with N. mediterraneum’.
2. L 109-125, L 200-201, the content could be clearly showed in Table. It is better for the understanding. For example, Table S1 could move to the manuscript and add detail information of those strains.
3. In Table 1, what is Average inoculation point? Please explain it. Is that the ‘girdling index ’?Please use the name of N. stellenboschiana for the present isolates in Table 1.
4. Many species are not in italic, L248, 254… Please check it in the whole manuscript.
5. Figure legends should show in detail (Fig. 1-2). Why not showing the colony feature of OL 438 which is different from the other three isolates based on the phylogenetic tree?
6. There are many figures in the manuscript. Some figures could be combined because of the similar information. For example Figure 7 and Figure 8.

Reviewer 3 Report
The manuscript is about the investigation into the dieback syndroms of olive trees. The authors report on the identification of a new fungal pathogen namely Neofusicoccum stellenboschiana, isolated at different sites in Central and Southern Italy. N. stellenboschiana is a rather newly identified pathogen, first isolated in South Africa and in the last years also found on different trees and shrubs around the Mediterranean. Here, the authors study 4 isolates for morphological, phenological traits and compare genomic traits. By this, it is possible to attribute the isolates taxonomically to the species N. stellenboschiana. In very complete inoculation studies, on young trees and on twigs, the authors characterize the pathogenicity and the aggressiveness of the isolates, completing, by this, Koch's postulate. The studiy is completed with the comparison of the pathogenicity with the known olive dieback pathogen N. mediterranea.
Overall, the study raises important and significant questions that are answered in a scientifically sound and clear manner. However, the manuscript bears a series of weaknesses and errors in its scientific conception, wording and English language.
